# Nanotechnology Innovations to Enhance the Therapeutic Efficacy of Quercetin

**DOI:** 10.3390/nano11102658

**Published:** 2021-10-09

**Authors:** Rúben G. R. Pinheiro, Marina Pinheiro, Ana Rute Neves

**Affiliations:** 1LAQV, REQUIMTE, Departamento de Ciências Químicas, Faculdade de Farmácia, Universidade do Porto, 4050-313 Porto, Portugal; poloaquatico_5@hotmail.com (R.G.R.P.); mpinheiro@med.uminho.pt (M.P.); 2CQM—Centro de Química da Madeira, Universidade da Madeira, Campus da Penteada, 9020-105 Funchal, Portugal

**Keywords:** Alzheimer prevention, cancer prevention, drug delivery, nanomedicine, natural compounds, nutraceutics

## Abstract

Quercetin is a flavonol present in many vegetables and fruits. Generally, quercetin can be found in aglycone and glycoside forms, mainly in leaves. The absorption of this compound occurs in the large and small intestine, where it suffers glucuronidation, sulfidation, and methylation to improve hydrophilicity. After metabolization, which occurs mainly in the gut, it is distributed throughout the whole organism and is excreted by feces, urine, and exhalation of carbon dioxide. Despite its in vitro cytotoxicity effects, in vivo studies with animal models ensure its safety. This compound can protect against cancer, cardiovascular diseases, chronic inflammation, oxidative stress, and neurodegenerative diseases due to its radical scavenging and anti-inflammatory properties. However, its poor bioavailability dampens the potential beneficial effects of this flavonoid. In that sense, many types of nanocarriers have been developed to improve quercetin solubility, as well as to design tissue-specific delivery systems. All these studies manage to improve the bioavailability of quercetin, allowing it to increase its concentration in the desired places. Collectively, quercetin can become a promising compound if nanotechnology is employed as a tool to enhance its therapeutic efficacy.

## 1. Introduction

### 1.1. Quercetin as a Promising Compound

Quercetin (3,3′,4′,5,7-pentahydroxyflavone) is a flavonol that was isolated and identified for the first time by Szent-Gyorgyi in 1936 (Figure 1) [1]. In the past, and similarly to what happened with other flavonoids, quercetin was classified as a vitamin shown to be important in maintaining capillary wall integrity and capillary resistance, as well as antihypertensive and antiarrhythmic activity. In subsequent studies, it was proven to have anti-inflammatory and antiallergic properties, hypocholesterolemic activity, platelet and mast cell stabilization, antihepatotoxic activity, and antitumor action [2,3]. Lastly, more in depth analysis demonstrated that the beneficial effects of quercetin were related with its antioxidant (free-radical scavenging) and anti-inflammatory activity [4]. Additionally, some reports also attest its neuroprotective properties. Thus, quercetin emerged as a promising candidate for the treatment of many diseases, such as cancer and neurodegenerative diseases. Nevertheless, in order to explore the potentialities of this flavanol, it is necessary to increase its bioavailability that is very reduced due to its hydrophobicity.

Nanotechnology is an appealing option to overcome this limitation since it can increase quercetin bioavailability at the desired site of action and, consequently, improve the therapeutic effectiveness. Hence, various kinds of NPs, namely liposomes, lipid nanoparticles, polymeric nanoparticles, cyclodextrins, silica nanoparticles, magnetic nanoparticles and gold nanoparticles, have been used to tackle this issue. Despite the fact that some authors have covered specific aspects of quercetin delivery, such as a particular class of nanosystems (polymeric vehicles) or medical applications (tumor therapy) [5,6], a comprehensive review summarizing all the data available regarding nanovehicles used for quercetin delivery is still needed. Intending to fill this gap, we aim to summarize the current literature regarding quercetin (e.g., chemical structure, natural sources, biosynthesis, pharmacokinetics, and health benefits) and compile an extensive analysis of how nanotechnology have been employed to maximize the therapeutic potential of this flavanol in its wide range of medical applications.

### 1.2. Natural Sources of Quercetin and Biosynthesis

Quercetin is one of the most abundant flavonoids in vegetables and fruits, being found mainly in onions, chilis, berries, and apples (Table 1) [7,8,9]. In leaves, it appears predominantly as aglycones or glycosides (3-position or/and 4′-position), with glucose being the most common sugar group. Nonetheless, lactose and rhamnose can also bind to phenolic groups of quercetin [7]. It is important to mention that, in onion, the best quercetin natural source, quercetin-4′-O-*β*-glucoside and quercetin-3,4′-O-*β*-diglucoside represent 80% of the total content of quercetin flavonoid types, being present in the highest quantities in the 28 vegetables and 9 fruits studied [10]. In a different study, quercetin was detected in all 25 eatable berries studied and the highest concentration was found in bog whortleberry (158 mg/kg, fresh weight) [11]. In a different work, quercetin-3-O-*β*-glucoside was discovered in a considerable amount in apple, pear peels, and Hypericum perforatum leaves or flowers [7,12]. Moreover, other studies have shown that black tea, red wine, and various fruit juices are also good alternatives as natural sources for quercetin [13].

The biosynthesis of quercetin shares almost the same steps in terms of metabolic pathway as other flavonoids. The three-ring structure (A, B, and C) with a diphenyl propane skeleton (C-6-C-3-C-6) is a hallmark of quercetin (Figure 2) [3]. The A and B are benzene rings linked by oxygen containing pyrene ring (C) [14]. The A ring is biosynthesized by the condensation of three moles of malonyl-coenzyme A (CoA) originated from the metabolism of glucose [3]. The B and C rings are also derived from glucose metabolism through the shikimic acid pathway to produce cinnamic acid and its reduced product, coumaric acid [3]. As CoA derivatives, this C-9 p-coumaroyl-CoA condenses with three moles of C-3 malonyl-CoA to form a C-15 chalcone. In the final step of the biosynthesis pathway, the ring closes and the hydration gives rise to quercetin (Figure 2) [15].

### 1.3. Chemical Structure and Properties of Quercetin

The IUPAC name of quercetin is 2-(3,4-dihydroxyphenyl)-3,5,7-trihydroxy-4H-chromen-4-one. Quercetin is composed of two benzene rings and one oxygen containing pyrene ring (Figure 1) [3]. Quercetin has amphipathic behavior due to the phenyl rings (hydrophobic part) and the hydroxyl groups (hydrophilic part) [16,17]. Nevertheless, quercetin presents low water solubility. There is some controversy in the literature concerning its solubility value; however, most studies indicate approximately 0.01 mg/mL (25 °C) [18,19,20]. Another important parameter while working with this flavonoid is its photodegradation. A photostability study has been conducted with quercetin alcoholic solutions, and the results revealed the appearance of some products of degradation measured by spectrophotometry under the exposure to UVB and UVA radiation, indicating the degradation of this polyphenol compound [21]. Another essential property relates with quercetin stability at different pH values. One study has demonstrated that quercetin is degraded at weak basic pH 8, while, at pH 5, almost 75% of quercetin remains in solution, indicating that quercetin is more stable in a protic medium [22]. Furthermore, a magnetic study of quercetin indicates that this compound displays diamagnetic properties [16]. This feature shows that valence electrons are paired, and, consequently, the compound seems to be stable. Still, the study of LUMO (lowest unoccupied molecular orbital) and HOMO (highest occupied molecular orbital) showed a little energy gap between those orbitals, indicating that an electron can transit from LUMO to HOMO orbitals and, by this way, can easily react [16]. Additionally, the potential ionization study points out to modest requirements in terms of energy to take away an electron from the valence orbital [16]. These two properties may explain the antioxidant capacity of quercetin.

## 2. Pharmacokinetics of Quercetin

### 2.1. Absorption, Distribution, Metabolism, and Elimination

Quercetin is a highly hydrophobic compound, so, when it reaches the small intestine, it can be absorbed by the epithelial cells, traversing through cellular membranes (phospholipid bilayers) by a simple diffusion pathway [23]. Chen et al. performed some absorption experiments in Sprague–Dawley rats and found that almost 60% of quercetin orally administered was absorbed [24]. This result was comparable to the work developed by Walle et al., where 53% of quercetin administered was absorbed [25]. Inside the enterocytes, the compound suffers glucuronidation and sulfidation at one of the hydroxyl groups by glucuronosyltransferases and sulfotransferases (phase II enzymes) in order to confer hydrophilicity [26,27,28,29]. Quercetin can also be O-methylated, primarily resulting in the formation of 3′-O-methylquercetin (isorhamnetin) and, to a smaller extent, 4′-O-methylquercetin (tamaraxetin) [28,29]. When quercetin is conjugated with sugars, the first step is to remove this group by β-glucosidase present in enterocytes and intestinal flora, and, after that, the quercetin can be conjugated as previously mentioned [26]. However, some molecules of quercetin can enter the circulation without suffering any conjugation. These molecules will reach the liver through the portal vein, and metabolization by glucuronosyltransferases and sulfotransferases will occur there, which are largely expressed enzymes [30]. Moreover, the catechol-O-methyltransferase (COMT) enzymes present in the liver and kidney can methylate quercetin [29,31,32]. If quercetin conjugates are excreted for bile, they flow through small intestine until they reach the hindgut where they can be hydrolyzed by the β-glucuronidase and sulfatase activities of the microflora, which allows the enterohepatic cycling, increasing the circulation time [30]. Although the liver is the critical organ for quercetin metabolization, a study showed that 90% of quercetin absorbed was metabolized in the gut [24]. All these enzymatic changes transform quercetin in a more soluble compound, allowing its blood circulation freely or bound to blood proteins, such as albumin [33,34]. The entrance of quercetin in blood circulation will result in its tissue distribution. It is also important to refer to the fact that quercetin can be absorbed from the gastrointestinal tract to the lymphatic system [28]. The regular ingestion of the compound will lead to the accumulation in many organs (i.e., lung, kidney, thymus, heart, liver) with the highest concentrations of quercetin and its methylated derivatives, particularly isorhamnetin, found in the pulmonary tissue [35]. Once ingested, quercetin remains in the organism for an extended period (20–72 h), most likely due to its enterohepatic recycling [25]. Nevertheless, quercetin can be degraded by microflora in the colon in phenolic acids and carbon dioxide, which is expelled in breath [36]. Contrarily, the phenolic acids can be excreted in feces. In addition, some quercetin may be also eliminated in urine [37,38,39]. All these routes are valid for quercetin elimination. In accordance to that, a study conducted by Ueno et al. showed that quercetin was excreted as expired CO_2_ (35%), or via the feces (45%) and urine (10%) as glucuronide or sulfate conjugates following oral administration [40]. However, in a more recent study, only a small amount of absorbed quercetin was eliminated in urine (3.3–5.7%) and feces (0.21–4.6%) [25]. The majority of quercetin was eliminated under carbon dioxide (41.8–63.9%) [25].

### 2.2. Toxicity of Quercetin

In terms of toxicity, there are still some contradictory results. The in vitro studies revealed that quercetin induces SOS activity and reverse mutations and DNA single strand breaks in bacteria (Salmonella typhimurium strains and Escherichia coli) and in eukaryotic cells, including yeast cells, at relatively high concentrations (up to 10 mg/incubation mixture) in the last case [41,42,43,44]. In hamster and mouse cells and human lymphocytes, harmful effects were reported as chromosomal aberrations, DNA single strand breaks, and micronucleus formation [45,46]. However, all these studies were not conducted in in vivo models. To further explore this issue, mice and rats were orally administered with quercetin and did not exhibit any significant changes in several mutagenicity/genotoxicity endpoints, i.e., micronuclei, chromosomal aberrations, sister chromatid exchange, unscheduled DNA synthesis, and alkali-labile DNA damage [46,47,48]. Based on the toxicity founded by in vitro assays, the carcinogenicity of quercetin was questioned using animal models. F344 rats receiving quercetin as 0.1% of the diet (50 mg/kg body weight/day) for 540 days did not differ from the control rats in terms of tumor incidence, except for lung adenoma and one jejunal adenocarcinoma [49]. In order to test the association between quercetin and lung cancer, quercetin was administered in the diet of A/JJms mice at 5% (7500 mg/kg body weight/day) for 23 weeks, and this compound did not induce a significant difference in the incidence of lung tumors, discarding the possibility of quercetin to induce lung cancer [50]. Therefore, all studies conducted in vivo with animal models indicate that quercetin is a safe compound.

## 3. Health Benefits of Quercetin

### 3.1. Cancer Chemoprevention

Quercetin has been reported as a cancer chemopreventive compound. This feature is associated to its ability to inhibit carcinogenesis via antimutagenic and antioxidant activity, anti-inflammatory mechanisms, modulation of signal transduction pathways, and apoptosis-inducing and anti-proliferative activity [51]. The Aryl hydrocarbon receptor (AhR), which binds to polycyclic aromatic hydrocarbons (PAHs), and halogenated aromatic hydrocarbons (HAHs) can activate the expression of CYP1A1, CYP1A2, and CYP1B1; consequently, these enzymes have the capacity to activate procarcinogens resulting in lung cancer [52]. Quercetin seems to naturally bind to AhR, being capable of preventing this signaling cascade and, consequently, cancer [52,53]. Quercetin has also the capacity to induce the expression of death receptor 5 (DR5) in lung cancer cells, which binds to TNF-related apoptosis-inducing ligand (TRAIL) and promotes apoptosis [54]. Moreover, quercetin can interfere with ErbB signaling pathway, reducing the expression of ErbB2 and ErbB3 in HT-29 colon cancer cells resulting in the inhibition of cell growth and the induction of apoptosis [55]. Another critical example relates to prostate cancer prevention. Quercetin demonstrates the capacity to reduce the expression of androgen receptor (AR) in human prostate cancer cell lines, LNCaP, and/or LAPC-4, slowing down cancer progression [56]. Overall, these studies clearly show the capacity of quercetin not only to prevent cancer but also to impair its progression.

### 3.2. Cardiovascular Protection

Quercetin can protect the cardiovascular system by multiple pathways. Galindo et al. conducted a study with rats and showed that regular intake of quercetin reduces the systolic blood pressure, normalizes the heart rate, reduces heart hypertrophy, and allows aortic relaxation by increasing nitric oxide and reducing some subunits expression of NADPH oxidase [57,58]. Furthermore, quercetin can activate fibrinolytic proteins (t-PA, u-PA, PAI-1, u-PAR, and Annexin-II) in mice, which disrupt fibrin clots in blood vessels, contributing to eliminate the thrombi and, consequently, lowering the risk of coronary heart diseases (CHD) [59]. Despite several studies showing the decrease of blood pressure in rats, there is the need to transpose these studies into humans. With that in mind, Edwards et al. proved that quercetin can lower blood pressure in stage 1 hypertensive patients, suggesting that the same principle also applies to humans [60].

### 3.3. Anti-Inflammatory Action

Currently, many studies point out to the anti-inflammatory capacity of this polyphenol compound. Mamani-Matsuda and colleagues have worked with rat models of arthritis, which correlates well to what happens in humans in terms of macrophage markers, and demonstrated that quercetin reduced the production of nitric oxide (NO), tumor necrosis factor (TNF-α), monocyte chemoattractant protein 1 (MCP-1), and interleukin 1 beta (IL-1β), which are the primary inflammatory and pro-arthritic mediators of macrophages [61]. Additionally, Rogerio et al. used murine models of asthma to prove the capacity of quercetin to lower the number of white blood cells and eosinophil in the bronchoalveolar lavage fluid, blood, and lung parenchyma [62]. Besides these studies in animal models, the anti-inflammatory ability of quercetin was also tested in human cells. Human mast cell lines were stimulated with phorbol 12-myristate 13-acetate (PMA) and calcium ionophore. Quercetin decreased the gene expression and production of TNF-α, interleukin-1β, IL-6, and IL-8 by reducing the activation of NF-kB and p38 mitogen-activated protein kinase [63]. In another study with human mast cells, Kimata et al. demonstrated the capacity of quercetin to impair the release of histamine, leukotrienes, prostaglandin D2, and granulocyte macrophage-colony stimulating factor [64]. Collectively, these studies gather strong evidences of the anti-inflammatory capacity of quercetin.

### 3.4. Antioxidant Activity

Compelling evidence shows the antioxidant activity of quercetin. This natural compound participates in many protective mechanisms, such as scavenging reactive oxygen species (ROS) and preventing ROS formation by chelating transition metal ions, such as iron and copper [65,66]. The radical scavenging ability of this flavonol relies on its chemical structure, particularly the hydroxyl (-OH) substitutions and the catechol-type B-ring [67]. Many studies attest this capacity, namely the works developed by Kim and Jang, who showed that quercetin had the ability to protect against the oxidative stress provoked by hydrogen peroxide in HepG2 cell line [68]. Moreover, Sánchez et al. proved that quercetin downregulates NADPH oxidase, increases endothelial nitric oxide synthase (eNOS) activity, and prevents endothelial dysfunction in spontaneously hypertensive rats, also emphasizing its cardiovascular protection through the oxidative stress reduction [58]. Besides that, Chen et al. demonstrated the inhibition of iNOS gene expression by quercetin in mouse BV-2 microglia [69]. Therefore, the literature provides several evidence regarding quercetin antioxidant activity.

### 3.5. Neuroprotection Effects

Several studies have focused their attention in the neuroprotection activity of quercetin. In fact, it can protect nerve cells from oxidative stress, increasing the survival of neurons [70,71,72]. Simultaneously, it can induce neuron differentiation contributing to maintain the balance of neuron number [73]. Not only to prevent but also to the treatment of neurodegenerative diseases, quercetin can be an interesting natural compound to attenuate the progressive degradation and loss of neuron cells. In this context, quercetin may play an important role in the attenuation of the massive oxidative stress caused by the accumulation of aggregates. For example, in the case of Alzheimer’s disease, Ansari et al. showed that quercetin was capable of reducing protein oxidation, lipid peroxidation, and apoptosis caused by amyloid beta-protein, Aβ(1–42), in primary hippocampal cultures [74]. Moreover, this flavonol has also shown the ability to inhibit the fibril formation of Aβ in a study conducted by Kim et al. [75]. Besides that, quercetin shows a varied spectrum of action regarding brain protection, such as increased mitochondrial biogenesis [76]. Indeed, this is very important because impaired mitochondrial activity seems to be correlated with neurodegenerative diseases [51]. Simultaneously, this natural compound is a potent anti-inflammatory and, therefore, can reduce the expression of proinflammatory molecules, contributing to reduction of the damage associated with this destructive inflammation process [77]. Overall, these studies show that quercetin might contribute to the health of the brain by preventing neurodegenerative diseases, as well as attenuating their deleterious effects.

## 4. Nanoparticles for Quercetin Delivery

Today, the challenge in drug delivery field consists of transporting drugs to their target places, allowing to increase compounds bioavailability, thereby lowering the amount of administered substances and, consequently, minimizing their side-effects while enhancing their therapeutic efficacy. This question is particularly important while dealing with hydrophobic molecules, as it is critical to design the best vehicles to increase their solubility and bioavailability in order to reach their target areas. Quercetin is one of these examples, and several studies reported in the literature devote significant attention to the development of several nanotechnological approaches which try to establish the best strategy to encapsulate and deliver quercetin for different applications. Table 2 summarizes some parameters that characterize different quercetin delivery systems, namely their composition, particle size, zeta potential, entrapment efficiency, administration route, and in vitro/in vivo results.

### 4.1. Liposomes

Liposomes can be described as nanocarriers which mimic the cellular phospholipid bilayers. The phospholipids used to produce liposomes have hydrophilic and hydrophobic portions allowing the spontaneous formation of spherical lipid bilayers in water formulations. The type of lipids chosen influences the properties of liposomes, such as charge, size, rigidity, etc. [118]. Liposomes composed of natural phospholipids are biologically inert and weakly immunogenic, which is crucial to biological applications [119]. Taking in consideration these advantages, liposomes have been used to encapsulate quercetin for many purposes [78,79,80,81,82,83,84,120]. Liu et al. have developed liposomes composed of phosphatidylcholine, cholesterol, and tween 80 to encapsulate quercetin in order to protect the hairless skin of Kun Ming mice against photodamage provoked by UVB [82]. Shaji et al. used multi-lamellar vesicles (MLV) with phosphatidylcholine and cholesterol in ratio of 9:1 to encapsulate quercetin and showed hepatoprotective activity in rats [79]. Thinking of tumoral therapy, Yuan et al. worked with a tumor-bearing mice model using lecithin/cholesterol/PEG4000/quercetin in 13:4:1:6 ratio and showed tumors growth inhibition [80]. Long et al. used PEGylated liposomes composed of lecithin and cholesterol to encapsulate quercetin, showing anti-tumor and anti-angiogenesis properties in ovarian cancer mouse model [81]. Interestingly, the delivery of quercetin in liposomes might be used to potentiate the drugs used to treat cancer. With this in mind, Wong et al. used liposomes to encapsulate vincristine and quercetin for treatment in a trastuzumab-insensitive breast tumor xenograft model [83]. This formulation was capable of increasing the circulation time of both drugs in plasma and inhibiting tumor growth [83]. For brain applications, Priprem et al. used quercetin liposomes in rats, with a mixture of egg phosphatidylcholine, cholesterol, and quercetin (2:1:1), dispersed in 50% polyethylene glycol in water. They managed to reduce the anxiety and verified cognitive-enhancing in rats [78]. At the same time, Phachonpai et al. also developed egg phosphatidylcholine/cholesterol liposomes to encapsulate quercetin and showed promising results via intracerebroventricular route administration, reducing degeneration of cholinergic neurons in hippocampus [120]. Finally, in human cell lines, Goniotaki et al. demonstrated that quercetin encapsulated in egg phosphatidylcholine liposomes can inhibit the growth of many human cancer cells [84].

### 4.2. Lipid Nanoparticles

Lipid nanoparticles can be classified as nanostructured lipid carriers (NLC) and solid lipid nanoparticles (SLN) [121]. The last one is composed of one or more solid lipids, which form a solid matrix, being an excellent vehicle for drug delivery because of their physical stability, protection of the incorporated drug from degradation, controlled release and low cytotoxicity [122]. Despite these advantages, their lipid matrix may suffer recrystallization while stored and form more perfect matrices (β-modifications) that can prematurely release the encapsulated compound [123]. In this context, NLC have been developed to overcome this problem because they blend liquid lipids with solid ones, creating an imperfect structure with more cavities and capacity to encapsulate drugs and avoiding premature release of the encapsulated compounds [123]. Several approaches using lipid nanoparticles have been developed in order to increase quercetin bioavailability and target specific places [86,87,88,89,90]. Li et al. produced SLN with glyceryl monostearate and soya lecithin, registering increased quercetin gastrointestinal absorption in rats [86]. However, it is also feasible the topical administration of quercetin in the skin to protect against oxidative stress caused by multiple factors (radiation, stress, among others). Bose et al. have performed some permeation studies using Franz diffusion cells with human skin and SLN composed of precirol and compritol in 3:2 ratio increased the content of quercetin inside skin, demonstrating that the lipid nanoparticles have great capacities as nanocarriers for topical delivery [88]. This idea was confirmed by Chen-yu et al. using glyceryl monostearate, stearic acid, and media chain triglyceride to prepare NLC for topical administration in ear edema-induced rats. The results showed a suppression of edema in the animals [87]. This capacity to protect against oxidative stress can be used in cancer therapy because cancer formation and progression are frequently associated with multiple mutations that can be caused by ROS. In a study conducted by Sun et al., it was possible to induce apoptosis of MCF-7 and MDA-MB-231 breast cancer cells using NLC composed of 2.7% quercetin, 9.4% soy lecithin, 23.6% glyceryltridecanoate, 6.7% glyceryl tripalmitate, 13.4% vitamin E acetate, and 44.2% Kolliphor HS15 [89]. Besides that, taking advantage of its neuroprotection properties, Dhawan et al. encapsulated quercetin in SLN composed of compritol and tween 80 [90]. They tested this formulation in rats chronically administered with aluminum chloride which causes an oxidative stress responsible for brain damaged [90]. The results showed that quercetin-loaded SLN ameliorated memory retention in rats with aluminum-induced dementia compared to quercetin alone and empty nanoparticles, indicating that this nanosystem may be efficient to target the brain [90]. Besides that, our team has also developed transferrin-functionalized and RVG-29-fuctionalized SLN and NLC made of cetyl palmitate, mygliol-812, and tween 80 to encapsulate quercetin for neuroprotection. The results showed that both types of functionalization enhanced the permeability of quercetin through hCMEC/D3 cells monolayers, and these nanoparticles seemed to be suitable for brain applications due to inhibition of amyloid-beta aggregation [92,93].

### 4.3. Polymeric Nanoparticles

Polymeric nanoparticles are made of biodegradable polymers which may offer multiple advantages, such as being stable in blood, biodegradable, non-toxic, nonthrombogenic, nonimmunogenic, and noninflammatory, not activating neutrophils, avoiding reticuloendothelial system, and being applicable to various molecules, such as drugs, proteins, peptides, or nucleic acids [124,125]. The versatility of these nanoparticles is based in the capacity to select the most appropriate polymer to the desired application. In this way, these nanomedicines have been used in many approaches [94,95,96,97,98,99,100,101]. Kumari et al. synthesized polylactic acid (PLA) nanoparticles with high encapsulation efficiency and controlled release of quercetin, making them promising for new therapy approaches [94]. Khoee et al. used methacrylated poly(lactic-co-glycolic acid) (mPLGA) as a lipophilic domain, acrylated methoxy poly(ethylene glycol) (aMPEG) as hydrophilic part and N-2-[(tert-butoxycarbonyl)amino] ethyl methacrylamide (Boc-AEMA) as pH-responsive part. They have proven the capacity of these polymeric nanoparticles to release their content in acidic environment, showing that they might be suitable for cancer therapy due to the acidic environment of tumor site [95]. El-Gogary et al. produced PEGylated PLGA nanoparticles conjugated with folic acid and demonstrated the capacity of this quercetin-loaded nanosystem to increase the apoptosis in HeLa cells compared to quercetin alone [96]. At the same time, the tumor uptake was confirmed in injected tumor-bearing mice [96]. Finally, Bishayee et al. used gold-quercetin into PLGA nanoparticles as a special system to escape the immune recognition [97]. The results showed that these nanoparticles had the ability to control proliferation and induce apoptosis in hepatocarcinoma cells [97].

#### Biopolymeric Nanoparticles

Interestingly, the sub-class of biopolymers are particularly attractive since they are non-toxic, biodegradable and biocompatible. This feature is majorly attributed to its monomer composition consisting essentially of nucleic acids, amino acids, and saccharides from sugars. Indeed, there are reports in the literature describing biopolymeric nanoparticles to deliver quercetin. In one of these studies, Wang et al. developed zein and dextran sulfate sodium binary complex to deliver quercetin. This nanovehicle exhibited almost no toxicity, as well as a sustained released, making it a promising candidate for future applications [104]. In another study, Lozano-Pérez et al. synthesized silk fibroin nanoparticles loaded with quercetin and observed a controlled released under the conditions of pH characteristic of intestinal fluid, which makes them suitable for gastrointestinal delivery [105]. In a follow up study, quercetin-loaded silk fibroin nanoparticles administered in a model mouse of colitis improved the pathological inflammation and preserved the normal crypt architecture, confirming the previously reported capacity for gastrointestinal delivery [106]. Another interesting biopolymer is chitosan, a cationic polysaccharide well known for its biological properties, making it a good candidate when developing drug delivery systems [126]. Pedro et al. developed pH-responsive amphiphilic chitosan nanoparticles to encapsulate quercetin for a sustained release in cancer therapy. The results indicated a good hemocompatibility of the nanoparticles and suggested they could inhibit cancer cell growth in vitro. Therefore, these chitosan nanoparticles may help in reducing the side effects upon systematic administration of quercetin as anti-cancer therapy [107].

### 4.4. Magnetic Nanoparticles

Magnetic nanoparticles are very promising nanosystems whose magnetic properties allow for guiding them to the target place, applying an external magnetic field [127]. It is also interesting to notice that, below a certain range of size (10–20 nm), magnetic nanoparticles behave as a giant paramagnetic atom with a fast response to applied magnetic fields with negligible remanence (residual magnetism) and coercivity (the field required to bring the magnetization to zero) [128]. The absence of residual magnetism is crucial because agglomeration can be prevented [128]. So far, there are not so many applications using magnetic nanoparticles to encapsulate quercetin; however, their use has now started to take its first steps [108,109,110,111]. For instance, magnetic nanoparticles are very promising for cancer therapy because external magnetic fields can direct them to the tumor site. Taking this in consideration and thinking on the cancer chemotherapeutic properties of quercetin, magnetic nanoparticles must be considered as potential and promising nanosystems for delivering quercetin to the tumor cells. In a preliminary study, Barreto et al. synthesized Fe_3_O_4_ nanoparticles and showed that these magnetic nanoparticles had a controlled releasing time, making this vehicle promising for cancer chemotherapy [108]. This study was followed by some studies of concrete applications of magnetic nanoparticles to deliver quercetin. Verma et al. used Fe_3_O_4_ magnetic core-shell nanoparticles protected against oxidation by PLGA and tested this formulation in the human lung carcinoma cell line A549 [109]. The results showed that quercetin-loaded PLGA-MNPs had no toxicity after injection in mice, and, at the same time, they were able to reduce the number of A549 viable cells, demonstrating anti-cancer activity [109]. In another study conducted by Kumar et al., quercetin superparamagnetic Fe_3_O_4_ nanoparticles were tested in vitro to analyze the effects in breast cancer cell lines [110]. Fluorescent microscopy demonstrated changes in cellular morphology of MCF7-treated cells, indicating cytotoxicity for cancer cells and consequently potential for cancer therapy [110]. Additionally, for targeting brain cancer, Akal et al. designed superparamagnetic iron oxide (SPION), which was functionalized with APTES ((3-aminopropyl) triethoxysilane), polyethylene glycol (PEG), and folic acid [111]. Folic acid was strategically used in order to target brain adenocarcinoma cells (U87), which overexpress folic acid receptors [111]. Furthermore, the MTT analysis after cellular uptake of SPION loaded with quercetin demonstrated decreased cancer cell viability [111]. Together, these results showed clear evidence that magnetic nanoparticles are suitable vehicles to deliver quercetin and to be used for cancer therapy.

### 4.5. Mesoporous silica Nanoparticles

Mesoporous silica nanoparticles (MSN) are becoming an appealing strategy since they are biocompatible, stable, have a tunable size pore, high drug encapsulation, slow drug release, and are suitable for being functionalized [129,130,131,132,133]. In the literature, there are some works using MSN as vehicles for quercetin delivery [112,113]. For example, Sapino et al. used MSN functionalized with aminopropyl for topical delivery of quercetin [112]. The results have shown that MSN increase the penetration of quercetin in the skin and, at the same time, inhibit the proliferation of R8 human melanoma cells [112]. In another work, MSN functionalized with folate, for targeting breast cancer cells, managed to induce cell cycle arrest and apoptosis in breast cancer cells through the regulation of Akt and Bax signaling pathways [113].

### 4.6. Cyclodextrins

Cyclodextrins (CDs) resemble a truncated cone with a hydrophobic cavity and a hydrophilic surface. In the cavity, it is possible to accommodate lipophilic drugs and deliver them to their target place [134]. Chemically, CDs are cyclic oligosaccharides with six, seven, or eight glucose residues linked by glycosidic bonds [135]. There are a few applications of CDs as nanodelivey systems to encapsulate quercetin [114,115,116,117]. The inclusion of the quercetin inside *β*-cyclodextrin (*β*-CD), hydroxypropyl-*β*-cyclodextrin (HP-*β*-CD), and sulfobutyl ether-*β*-cyclodextrin (SBE-*β*-CD) has been studied [114,115]. The results demonstrate that SBE-*β*-CD can encapsulate more quercetin than the other CDs tested and, at the same time, may increase its antioxidant properties [114,115]. Moreover, Aytac et al. used quercetin/*β*-cyclodextrin inclusion complex, demonstrating the slow release of quercetin [116]. All of these promising features as nanocarriers for quercetin were confirmed in a study conducted by Kale et al. [117]. They used ether-7*β*-cyclodextrin/quercetin complex in melanoma mouse models and verified the decrease of microvessels and, consequently, the reduction of tumor cell proliferation [117]. Although scarce, the studies here indicated show that CDs holds a great potential for future applications.

## 5. Clinical Trials and Marketed Formulations

The potentialities of quercetin are well documented in many in vitro and in vivo studies, as cited above. With this in mind, many clinical trials are being made to prove the realistic capacity of quercetin as a tool for new therapies (Table 3) [136]. Ninety-three clinical trials were founded related with quercetin [137]. In a generic view, most of the studies are centered in the United States of America. These clinical trials mainly explore the anticancer, antioxidant, anti-inflammatory, cardiovascular protection, and the anti-infectious properties of quercetin, including against SARS-CoV-2 [137,138]. Although many studies have been completed, only a few have published the results, perhaps due to inconclusive results. One of these studies was focused on the treatment of chronic obstructive pulmonary disease, where quercetin was able to decrease lung inflammation and prevent disease progression in a patient group of 40 to 65 years in age [139]. In the cancer area, quercetin is being studied as a therapeutic agent in colon, pancreas, and prostate cancer [140]. Moreover, some studies are concentrated on the capacity of quercetin to regulate blood pressure and the maintenance of an efficient cardiovascular system [141]. At last, other clinical trials are currently exploring the anti-inflammatory and antioxidant capacity of quercetin, for example, in the treatment of idiopathic pulmonary fibrosis and sarcoidosis [142]. None of the clinical trials mentioned used nanoparticles technology yet. However, compelling evidence demonstrate its benefit in terms of increased drug half-life and solubility, improved drug accumulation on the target site, and reduction of side effects. Due to the aforementioned beneficial properties, quercetin is being used in many market formulations (Table 4). In general, it is possible to find quercetin in many dietary supplements to complement the diet regimen with a potent antioxidant and anti-inflammatory molecule, especially as capsules or tablets [143]. At the same time, this flavonoid can be found in many cosmetics [144]. Brands are creating new formulas to fight the problem of aging [144]. Additionally, some make-up products also contain quercetin in their composition, as indicated in Table 4. Therefore, quercetin is widely used in many commercially available products.

## 6. Conclusions

The chemical reactivity of quercetin relies on its radical scavenging properties that enable reducing the oxidative stress of cells. At the same time, quercetin presents a strong anti-inflammatory capacity which, together with its antioxidant activity, gives rise to remarkable potentialities, such as cancer chemoprevention, cardiovascular protection, and neurodegeneration attenuation. However, this compound has low water solubility, chemical instability, rapid metabolism, and poor bioavailability, which compromises its therapeutic efficacy. Therefore, new approaches are necessary to take advantage of quercetin beneficial effects, while enhancing its potential application for treating and preventing several disorders. In this context, nanoparticles are particularly interesting when used to accomplish the task of delivering quercetin and enhancing its bioavailability. Furthermore, the capacity of functionalization with specific ligands which target specific organs or cells is also very important because one can increase quercetin concentration on the desired target site, while reducing side effects. Thinking of the beneficial effects of quercetin, a wide range of strategies that use nanoparticles have been developed to deliver quercetin in a specific and controlled way. The revised nanosystems described in the literature revealed promising approaches with a great capacity for quercetin encapsulation and controlled release. Moreover, several strategies of nanoparticle functionalization have allowed the tissue-specific delivery of quercetin to tumor microenvironments, as well as to increase its blood brain barrier permeation or the penetration within skin layers. However, there is a lack of clinical trials employing nanodelivery systems for quercetin administration, considering all the advantages mentioned in this review. Therefore, we identify here a potential area to be explored in future reviews, trying to understand why such promising nanosystems are not yet being widely used in clinical practice and how we can enhance their future use by taking advantage of the therapeutic properties of such a promising compound—quercetin. This strategy has great potential to be explored in a short period of time. Hence, the beneficial effects of quercetin may be further enhanced using nanotechnology, thereby helping to improve the current application of this compound with so great therapeutic potential.

## Figures and Tables

**Figure 1 nanomaterials-11-02658-f001:**
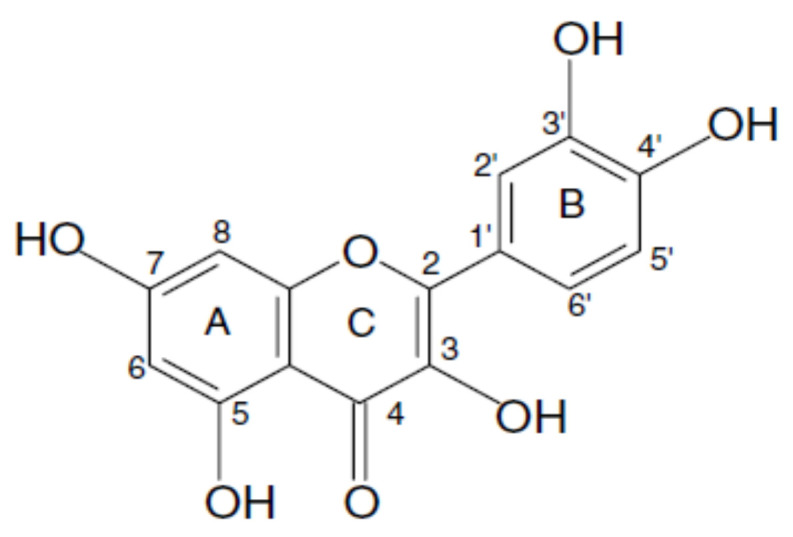
Chemical structure of quercetin.

**Figure 2 nanomaterials-11-02658-f002:**
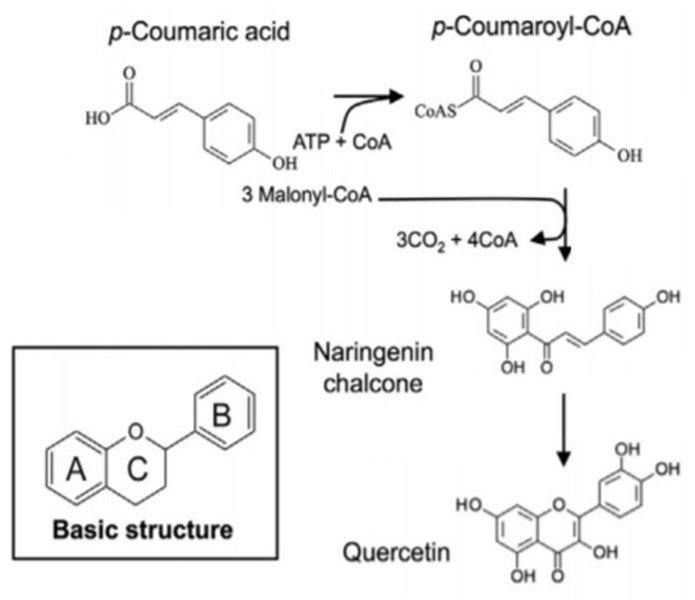
Schematic representation of quercetin biosynthesis.

**Table 1 nanomaterials-11-02658-t001:** Quercetin natural sources [1,2,3].

Sources	Total Quercetin (mg/Kg)
White onion bulbs	2604
Onion dry outer skin	960
Spring onion leaves	450
Chili powder	400
Bog whortleberry	158
Lingonberry	146
Cranberry	121
Kale	110
Chokeberry	89
Sweet rowan	85
Rowanberry	63
Sea buckthorn berry	62
Apples red delicious	58
Crowberry	56
Broccoli	30
Green beans	25
Apple peel	21
Tomato	13

**Table 2 nanomaterials-11-02658-t002:** Properties of different quercetin-loaded carriers.

NPs Type	Composition	Morphology	Particle Size (nm)	Zeta Potential (mV)	Encapsulation Efficiency (EE) (%)	Administration Route	In Vitro/In Vivo Results	Reference
**Liposomes**	PC/Chol; EPC/Chol/PEG; lecithin/Chol/PEG; ESM/Chol/PEG P90G/STA/Eudragit S100	spherical	90 to 450	+7 to −30	65 to 95	oral, intranasal, intravenous, topical	- anxiolytic and cognitive-enhancing effects in rats (in vivo); - hepatoprotective effects in rodents (in vivo); - attenuation of edema and inflammation in irradiated mice (in vivo); - inhibition of tumor angiogenesis, inhibition of tumor cell growth and induction of tumor cell apoptosis (cell lines and mice) (in vitro and in vivo); - protection against intestinal oxidative stress in intestinal cell lines (in vitro);	[78,79,80,81,82,83,84,85]
**Lipid nanoparticles**	GMS/soya lecithin/PEG; GMS/SA/MCT/soya lecithin; Compritol; GT/TG/soya lecithin SA/P-188; cetyl palmitate, mygliol-812 and tween 80	spherical	35 to 550	−10 to −35	75 to 95	oral, intravenous, topical, intraperitoneal	- enhancement of the oral absorption in rats (in vivo); - increase of drug penetration in skin and anti-oxidation and anti-inflammation effect (in vivo); - improvement of memory retention in rats with aluminum-induced dementia (in vivo); - induction of tumor cell apoptosis in breast cancer cell lines (in vitro). - improvements of cognition and memory impairments in zebrafish model of neurodegenerative disorders (in vivo). - inhibition of amyloid-beta aggregation and fibril formation in vitro, being suitable for brain applications (in vitro).	[86,87,88,89,90,91,92,93]
**Polymeric nanoparticles**	PLA; PLGA; PLGA/PEG/AEMA; PLGA/PEG/FA; PCL/TPGS; PLGA/TPGS Eudragit S100 L-CS/DEAE	spherical, bean-like shape	30 to 415	+42 to −40	40 to 98	intravenous, topical	- enhancement of cancer cell uptake in tumor-bearing mice (in vivo); - increase of cytotoxicity and induction of apoptosis in breast, colon and liver hepatocellular carcinoma cell lines (in vitro); - block/reduce of histological alterations in irradiated mice (in vivo); - enhancement of neuron cells viability due to inhibition of aβ-42 peptide neurotoxicity in cell lines (in vitro); - improvements of cognition and memory impairments in APP/PS1 mice model of Alzheimer’s disease (in vivo).	[94,95,96,97,98,99,100,101,102,103]
**Biopolymeric nanoparticles**	Zein-Dextran sulfate sodium; Skin Fibroin; Chitosan	spherical	140 to 300	−16 to −39; +14 to +30	20 to 78	oral, intravenous	- promising favorable and sustained delivery of quercetin (in vitro); - controlled release in the pH of intestinal fluid; suitable for gastrointestinal delivery (in vitro); - improving pathological inflammation and preserving normal crypt architecture in model mouse of colitis (in vivo); - pH-responsive sustained release for cancer therapy; reducing side effects upon systematic administration (in vitro).	[104,105,106,107]
**Magnetic nanoparticles**	Fe_3_O_4_; Fe_3_O_4_/E_137_S_18_E_137_; Fe_3_O_4_/APTS/PEG/FA; Fe_3_O_4_/PLGA	spherical	10 to 300	around +6	around 80	intranasal	- induction of cytotoxicity in human lung carcinoma, breast cancer and glioblastoma cell lines (in vitro).	[108,109,110,111]
**Mesoporous silica nanoparticles**	TEOS/APTS; TEOS/FA	spherical	200 to 250	−25 to +13	around 99	n.i.	- inhibition of tumor cell proliferation and induction of tumor cell apoptosis (cell lines) (in vitro).	[112,113]
**Cyclodextrins**	βCD; HP-βCD; SBE-βCD; SBE-7βCD	truncated cone; fibers	around 270	n.i.	n.i.	oral	- increase of quercetin solubility and photostability; - enhancement of quercetin antioxidant capacity (in vivo); - inhibition of tumor cell proliferation in cell lines (in vitro); - impairment of tumor growth in B16F10 mouse melanoma model (in vivo).	[114,115,116,117]

**Table 3 nanomaterials-11-02658-t003:** Quercetin clinical trials conducted around the world (reported at http://clinicaltrials.gov/, last access on 8 October 2021).

Country	Study	Condition	Intervention	Phase	Status	Results
U.S.—Michigan	Phase I Study to Determine the Safety of Quercetin in COPD Patients	Chronic Obstructive Pulmonary Disease	Quercetin	1	Completed with results	Decreased lung inflammation and prevented progression
U.S.—Massachusetts	Hypoxia-inducible Transcription Factor 1 (HIF-1) in Vascular Aging	Stroke; Problem of Aging	Quercetin	Early phase 1	completed	Not Indicated
U.S.—Ohio	Quercetin in Children with Fanconi Anemia; a Pilot Study	Fanconi Anemia	Quercetin	1	Recruiting	There are no results
U.S.—North Carolina	Can Quercetin Increase Claudin-4 and Improve Esophageal Barrier Function in GERD?	Gastroesophageal Reflux Disease (GERD); Acid Reflux; Reflux	quercetin	1	completed	Not indicated
U.S.—California	A Phase 1 Study of Quercetin in Patients with Hepatitis C	Chronic Hepatitis C	quercetin	1	completed	Not indicated
The Netherlands	The Effect of Quercetin on the Increased Inflammatory and Decreased Antioxidant Status in Sarcoidosis	Sarcoidosis	quercetin	-	completed	Not indicated
The Netherlands	Study on the Effects of Epicatechin and Quercetin Supplementation on Vascular Function and Blood Pressure in Untreated (Pre)Hypertensive Subjects	Hypertension Endothelial Dysfunction	Epicatechin; quercetin	-	completed	Not indicated
Islamic Republic of Iran	Effect of Quercetin in Prevention and Treatment of Chemotherapy Induced Oral Mucositis in Blood Dyscrasias	Chemotherapy Induced Oral Mucositis	oral quercetin capsules	1,2	completed	Not indicated
U.S.—New Mexico	Effect of Combined Exercise, Heat, and Quercetin Supplementation on Whole Body Stress Response	Heat Acclimation and Thermotolerance	Quercetin	-	completed	Not indicated
U.S.—California	Phase I Randomized, Double-Blind, Placebo-Controlled Two-Arm Study of Quercetin and Green Tea to Enhance the Bioavailability of Green Tea Polyphenols in Men Scheduled for Prostatectomy	Adenocarcinoma of the Prostate; Recurrent Prostate Cancer; Stage I Prostate Cancer; Stage IIA Prostate Cancer; Stage IIB Prostate Cancer; Stage III Prostate Cancer; Stage IV Prostate Cancer;	Dietary Supplement: green tea extract Drug: quercetin Other: placebo Procedure: therapeutic conventional surgery Other: laboratory biomarker analysis	1	Active, not recruiting	There are no results
U.S.—Massachusetts	Pharmacokinetic and Pharmacodynamic Study of Oral Quercetin and Isoquercetin in Healthy Adults and Patients with Elevated Anti-phospholipid Antibodies	Healthy	Drug: isoquercetin or quercetin	-	Active, not recruiting	There are no results
U.S.—Washington	Evaluation of Quercetin in Type 2 Diabetes: Impact on Glucose Tolerance and Postprandial Endothelial Function.	Diabetes Mellitus, Type 2	Dietary Supplement: Quercetin Drug: Acarbose Drug: placebo	2	completed	Not indicated
Islamic Republic of Iran	Therapeutic Effect of Quercetin and the Current Treatment of Erosive and Atrophic Oral Lichen Planus	Atrophic Oral Lichen Planus; Erosive Oral Lichen Planus	Drug: placebo Drug: Quercetin	1	unknow	There is no information
Germany	Clinical Trial on the Effectiveness of the Flavonoids Genistein and Quercetin in Men with Rising Prostate-specific Antigen	Primary Prevention of Prostate Cancer	Dietary Supplement: Quercetin supplement Dietary Supplement: Genistein supplement Dietary Supplement: Placebo	-	unknown	There is no information
U.S.—Texas	Prospective Open Labeled Pilot Trial of Quercetin in the Treatment and Prevention of Chemotherapy-induced Neuropathic Pain in Cancer Patients	Polyneuropathies and Other Disorders of the Peripheral Nervous System Chemotherapy Induced Neuropathic Pain	Behavioral: Questionnaires Drug: Quercetin	Early Phase 1	Not yet recruiting	There are no results
U.S.—Maryland	Inhibition of Intestinal Glucose Absorption by the Bioflavonoid Quercetin in the Obese and in Obese Type 2 Diabetics	Diabetes Mellitus; Obesity	Procedure: Oral glucose tolerance test; coadministration of 1 or 2 g of quercetin with glucose	2	recruiting	There are no results
United Kingdom	A Double-Blind, Cross-Over, Placebo-Controlled Study Evaluating the Effect of Quercetin 500 mg Tablets on Blood Uric Acid in Healthy Males	Hyperuricemia; Gout; Kidney Calculi; Diabetes; Cardiovascular Disease	Dietary Supplement: Treatment Dietary Supplement: Control	Early Phase 1	completed	There are no results
U.S.—Pennsylvania	Advancing Niacin by Inhibiting FLUSHing: (ANTI-FLUSH)	Flushing	Dietary Supplement: Quercetin Dietary Supplement: Placebo	4	Completed, has results	Not indicated
Greece	Effects of the Anti-inflammatory Flavonoid Luteolin on Behavior in Children with Autism Spectrum Disorders	Autism Spectrum Disorders	Dietary Supplement: Luteolin, Quercetin and Rutin combined in a capsule	2	completed	There are no results
U.S.—Alabama	Nasal Potential Studies Utilizing CFTR Modulators (UAB Center for Clinical and Translational Science)	Cystic Fibrosis	Other: quercetin	2	completed	There are no results
U.S.—North Carolina	Targeted Removal of Pro-Inflammatory Cells: An Open Label Human Pilot Study in Idiopathic Pulmonary Fibrosis	Idiopathic Pulmonary Fibrosis (IPF)	Drug: Dasatinib + Quercetin	1	recruiting	There are no results
U.S.—Minnesota	Senescence, Frailty, and Mesenchymal Stem Cell Functionality in Chronic Kidney Disease: Effect of Senolytic Agents	Chronic Kidney Disease	Drug: Group 2: Dasatinib Drug: Group 2: Quercetin	2	recruiting	There are no results
U.S.—Minnesota	Hematopoietic Stem Cell Transplant Survivors Study (HTSS Study)	Stem Cell Transplant	Drug: Quercetin Other: Standard of Care—Observation Only Drug: Dasatinib	-	recruiting	There are no results
U.S.—Utah	A Double-Blind, Placebo-Controlled, Crossover Evaluation of a Grape Seed Extract and Quercetin Supplement (Provex CV) to Reduce Markers of Inflammatory Cytokines and Blood Pressure in Subjects with Metabolic Syndrome	Blood Pressure	Drug: Provex CV Other: placebo	1	Completed	There are no results
U.S.—New York	The Effect of Plant Phenolic Compounds on Human Colon Epithelial Cells	Colorectal Cancer	Dietary Supplement: curcumin Dietary Supplement: rutin Drug: quercetin Drug: sulindac	-	terminated	There are no results
U.S.—Massachusetts	Absorption Kinetics of Polyphenols in Angel’s Plant (Angelica Keiskei)	Oxidative Stress	Dietary Supplement: angel’s plant (Angelica keiskei) Dietary Supplement: Angel’s plant (Angelica keiskei)	Early Phase 1	unknown	There is no information
U.S.—Michigan	Three New Ideas to Protect Special Forces from the Stress of High Altitude	Mountain Sickness	Dietary Supplement: Quercetin Drug: Nifedipine extended release Drug: Methazolamide Drug: Metformin Drug: Placebo Drug: Nitrite	4	recruiting	There are no results
The Netherlands	The Inflammatory and Antioxidant Status in Pulmonary Sarcoidosis, Idiopathic Pulmonary Fibrosis and COPD: a Potential Role for Antioxidants	Interstitial Lung Diseases; Sarcoidosis; Idiopathic Pulmonary Fibrosis; COPD;	quercetin	-	completed	There are no results
Republic of Korea	Effect of Onion Peel Extract on Endothelial Function and Endothelial Progenitor Cells in Overweight and Obese Subjects	Healthy Overweight Obese	Drug: Placebo Drug: Onion peel extract	4	completed	There are no results
United Kingdom	The Impact of Isoquercetin and Aspirin on Platelet Function	Cardiovascular Disease	Drug: Vehicle control Drug: Isoquercetin Drug: Aspirin Drug: Isoquercetin plus Aspirin	-	withdrawn	There are no results
U.S.—California	Randomized, Placebo-controlled, Double-blind Phase II/III Trial of Oral Isoquercetin to Prevent Venous Thromboembolic Events in Cancer Patients.	Thromboembolism of Vein VTE in Colorectal Cancer; Thromboembolism of Vein in Pancreatic Cancer; Thromboembolism of Vein in Non-small Cell Lung Cancer	Drug: Isoquercetin	2,3	recruiting	There are no results
Germany	Pilot Study Evaluating Broccoli Sprouts in Advanced Pancreatic Cancer [POUDER Trial]	Pancreatic Ductal Adenocarcinoma	Dietary Supplement: Verum, broccoli sprout grain Dietary Supplement: placebo	-	recruiting	There are no results
-	An Open, Uncontrolled Trial in Healthy Volunteers to Explore the Plasma and Urinary Pharmacokinetics of a Single Oral Dose of 1800 mg Red Vine Leaf Extract (Antistax^®^)	Healthy	Drug: Red Vine Leaf Extract	1	completed	There are no results
U.S.—North Carolina	Influence of Ingesting a Flavonoid-rich Supplement on the Human Metabolome and Concentration of Urine Phenolics	Inflammation	Dietary Supplement: Flavonoid Dietary Supplement: Placebo	-	Active, not recruiting	There are no results
Italy	Isoquercetin as an Adjunct Therapy in Patients with Kidney Cancer Receiving First-line Sunitinib: a Phase I/II Trial	Renal Cell Carcinoma; Kidney Cancer	Drug: Sunitinib Drug: Isoquercetin Drug: Placebo	1,2	Recruiting	There are no results
U.S.—Illinois, New Jersey and Ohio	Open-Label Study to Evaluate the Effect of Elimune Capsules on Biomarkers in Patients with Plaque Psoriasis	Plaque Psoriasis	Dietary Supplement: Elimune Capsules	1	completed	There are no results
France	Randomized Controlled Double-blind Clinical Trial for the Effect of Yoghurts Enriched in XXS vs without XXS on the Evolution of Weight in Overweight Subjects Aged 50 to 65 Years.	Obesity Overweight	Dietary Supplement: yoghurts enriched with XXS Dietary Supplement: yoghurts non enriched with XXS	-	completed	There are no results
Brazil	Evaluation of Clinical Efficacy Capsules Containing Standardized Extract of Bauhinia Forficata (Pata- De-vaca) in Diabetic Patients	Diabetes	Drug: B. forficata Drug: Placebo	-	recruiting	There are no results
Egypt??	Effects of Oral Ginkgo Biloba Extract on Pregnancy Complicated by Asymmetrically Intrauterine Growth Restriction: a Double-blinded Randomized Placebo-controlled Trial	Intrauterine Growth Restriction (IUGR)	Drug: Ginkgo Biloba Extract Other: Placebo	2	completed	There are no results
Italy	Physiological Effects of New Polyphenol-enriched Foods in Healthy Humans	Healthy	Other: free curcumin Other: encapsulated curcumin Other: encapsulated curcumin + PQG Other: free cocoa polyphenol Other: encapsulated cocoa polyphenols Other: control nut cream Encapsulada a quercetina	-	Completed Has Results	There are no results
U.S.—Missouri	Can Fish Oil and Phytochemical Supplements Mimic Anti-Aging Effects of Calorie Restriction?	Healthy Volunteers	Dietary Supplement: Supplement	-	completed	There are no results
U.S.—Michigan	GRAPe Seed Extract and Ventriculovascular Investigation in Normal Ejection-Fraction Heart Failure	Diastolic Heart Failure; Hypertensive; Heart Disease; ;Heart Failure with Preserved Ejection Fraction; Hypertension; Oxidative Stress	Drug: grape seed extract (MegaNatural BP, Polyphenolics, Inc.)	1	Active, not recruiting	There are no results
U.S.—South Carolina	Effects of Short-term Curcumin and Multi-polyphenol Supplementation on the Anti-inflammatory Properties of HDL (PSI)	Inflammation; Atherosclerosis; Cardiovascular Disease	Dietary Supplement: PolyResveratrol Supplementation Dietary Supplement: Curcumin Supplementation	2	Recruiting	There are no results
Canada, Alberta	Orthomolecular Treatment as add-on Therapy for Childhood Asthma	Asthma	Dietary Supplement: Orthomolecular Therapy or Placebo Comparator	2	unknown	There are no results
U.S.—California	Effects of Plant Concentrate Blend on Oxidative Stress in Healthy Humans	Healthy	Dietary Supplement: AOX blend Dietary Supplement: Placebo	-	completed	There are no results
U.S.—Ohio	Phytochemical Release Rate from Black Raspberry Confections Alters Gene Expression and Chemical Profiles Relevant to Inhibition of Oral Carcinogenesis	Healthy Volunteers	Other: Fast release BRB confection Other: Intermed release BRBconfect Other: Prolong release BRB confect	1	Active, not recruiting	There are no results
Norway	Dietary Intervention in Stage III/IV Follicular Lymphoma. Impact on Markers of Cell Proliferation, Apoptosis, Host Immune Cell Infiltrate and Oxidative Stress	Follicular Lymphoma	Drug: Omega 3 fatty acids (EPA (eicosapentaenoic acid) and DHA (docosahexaenoic acid)) Drug: Selenium (L-Selenomethionine), Drug: Garlic extract (Allicin) Drug: Pomegranate juice (ellagic acid) Drug: Grape juice (resveratrol, quercetin) Drug: Green Tea (Epigallocathechin gallate	2	unknown	There are no results
U.S.—Florida	Open Label, Crossover, Pilot Study to Assess the Efficacy & Safety of Perispinal Admin. of Etanercept(Enbrel^®^) in Comb. w/Nutritional Supplements vs. Nutritional Supplements Alone in Subj. w/Mild to Mod. Alzheimer’s Disease Receiving Std. Care	Alzheimer’s Disease	Biological: Etanercept Dietary Supplement: Curcum. Luteol. Theaflav. Lip. Acid, FishOil, Quercet, Resveratr.	1	Completed	There are no results
U.S.—California	Proof of Concept Study to Evaluate the Effect of Oxidative Stress Response of Plant Concentrate Blends in Healthy Men	Healthy	Dietary Supplement: Plant concentrate A Dietary Supplement: Plant concentrate B Dietary Supplement: Plant concentrate C Dietary Supplement: Plant concentrate D	-	completed	There are no results
U.S.—Ohio	Quercetin chemoprevention for squamous cell carcinoma in patients with Fanconi anemia	Fanconi anemia Squamous cell carcinoma	Dietary supplement	Phase 2	Recruiting	There are no results
U.S.—Pennsylvania	Impact of quercetin on inflammatory and oxidative stress markers in COPD	Chronic obstructive pulmonary disease	Quercetin	Phase 1/Phase 2	Not yet recruiting	There are no results
Not indicated	Serum concentration and gene expression of sirtuin-1 and advanced glycation End-products in postmenopausal women with atherosclerotic coronary disease after administration of atorvastatin and supplementation with quercetin: Randomized trial sirtuin-1 and advanced glycation End-products in postmenopausal women with coronary disease	Coronary artery disease progression	Quercetin	Phase 3	Not yet recruiting	There are no results

**Table 4 nanomaterials-11-02658-t004:** Quercetin marketed formulations.

Brand	Dosage	Pharmaceutical Form	Application
Quercetin 500 mg (Nature’s Best)	500 mg	Tablets	Supplement
Rutin and vitamin c 500 mg (Nature’s Best)	50 mg	Tablets	supplement
Glucosamine Plus (Nature’s Best)	25 mg	Tablets	supplement
Glucosamine Gold (Nature’s Best)	20 mg	Tablets	supplement
Quercetin with bromelain (Biovea)	500 mg	Capsules	Vegan supplement
Neuroprotek Low Phenol (algonot)	70 mg	capsules	Dietary supplement
Neuroprotek (algonot)	70 mg	capsules	Dietary supplement
Algonot plus (algonot)	150 mg	capsules	Dietary supplement
ArthroSoft (algonot)	150 mg	capsules	Dietary supplement
CystoProtek (algonot)	150 mg	capsules	Dietary supplement
FibroProtek (algonot)	85 mg	capsules	Dietary supplement
ProstaProtek (algonot)	-	capsules	Dietary supplement
Quercetin (Natrol)	250 mg	capsules	Dietary supplement
Quercetin (Desert Harvest)	250 mg	capsules	Dietary supplements
Quercetin and vitamin (Vitacost)	250 mg	capsules	Dietary supplements
Aller-Aid with quercetin (Wild Harvest)	137,5 mg	capsules	Herbal supplement
Advan-c (Vitacost)	100 mg	capsules	Vitamin supplement
Quercetin and Bromelain (Vitacost)	250 mg	capsules	Dietary supplement
Quercetin (Jarrow formulas)	500 mg	capsules	Dietary supplement
Quercetin (solaray)	500 mg	capsules	Dietary supplement
Quercetin and Bromelain (Now)	400 mg	capsules	Vegan supplement
Activated quercetin complex (Vitacost)	333 mg	capsules	Dietary supplement
Quercetin complex (solgar)	250 mg	capsules	Dietary supplement
Mega quercetin (solaray)	600 mg	capsules	Dietary supplement
Quercetin strength (megaFood)	500 mg	capsules	Dietary supplement
Activated quercetin (source naturals	333 mg	capsules	Dietary supplement
Optimized quercetin LifeExtension	250 mg	capsules	Dietary supplement
Nettle quercetin (eclectic institute)	175 mg	capsules	Dietary supplement
Quercetin (natural factors)	235 mg	capsules	Dietary supplement
Quercetin (nature’s life)	400 mg	capsules	Antioxidant supplement
Quercetin (Pure encapsulations)	250 mg	capsules	Dietary supplement
Quercetin bioflavonoids (NutriCology)	50 mg	capsules	Dietary supplement
Quercetin (MRM)	500 mg	capsules	Dietary supplement
Quercetin (Kal)	1000 mg	tablets	Dietary supplement
Quercetin plus (Olympian labs)	500 mg	capsules	Dietary supplements
ALLER-C (Vital Nutrients)	250 mg	capsules	Dietary supplements
Quercetin Bromelain (Doctor’s BEST)	250 mg	capsules	Dietary supplement
Rose Plus Anti-Ageing Face (The organic Pharmacy)	n.i	Cream	anti-ageing cream
Flash Defense Anti-Pollution Mist (REN)	n.i	cream	skincare
Antiwrinkle Firming and Lifting serum (Korres)	n.i	serum	skincare
Tightening eye contour gel (Alchimie forever)	n.i	gel	Make up
Firming gel for neck and bust (Alchimie forever)	n.i	gel	skincare
Kantic calming cream (Alchimie forever)	n.i	cream	skincare
Rejuvenating eye balm (Alchimie forever)	n.i	cream	skincare
Antioxidant skin repair gel (Alchimie forever)	n.i	gel	skincare
Gentle cream cleanser (Alchimie forever)	n.i	cream	skincare
Resist advanced replenishing toner (Paula’s choice)	n.i	cream	skincare

## Data Availability

The data presented in this study are available on request from the corresponding author.

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
