# Peer review of "Nanotechnology Innovations to Enhance the Therapeutic Efficacy of Quercetin"

_nanomaterials, 2021, doi:10.3390/nano11102658_

Round 1

Reviewer 1 Report

The review entitled “ Nanotechnology innovations to enhance the therapeutic efficacy of quercetin” from  Rúben G. R. Pinheiro et al. is a well-structured review focused in the nanocarriers which have been recently developed in order to improve quercetin therapeutic efficacy, increasing their solubility as well as their tissue-specific delivery.

In my opinion, the review would gain in interest if the authors could provide/discuss/correct the following aspects:

  1. In the introduction, authors should contextualize the review in the field of nanotechnology, citing other previous similar reviews (For example: DOI: 10.1039/c5ra18896b or DOI:10.1039/D1FO00851J) and including a brief sentence justifying the necessity of the present revision.
  2. The section “4. Nanoparticles for quercetin delivery” must include a sub-section about the “Biopolymeric nanoparticles”, as good candidates for drug delivery in medicinal applications because they are constituted by highly versatile, non-toxic, biocompatible and biodegradable molecules, with an increase number of examples and outstanding applications. There are many examples in literature describing nanoformulations of quercetin with silk fibroin (10.1016/j.ijpharm.2016.12.046; 10.1016/j.ijpharm.2021.120935), zein (doi: 10.3389/fchem.2020.00662), or chitosan (10.3390/polym10030235) among others.
  3. The review presents a lack of schemes or figures presenting the wide variety of nanocarriers available for Quercetin transport and it would help to understand the different mechanisms of transport. Indeed, the figure 2 presents a lower resolution than figure 1. Please, check it.
  4. Please, edit the whole document according the journal requirements. The manuscript contain paragraphs with different text adjustment (for example in page 6) and check the page orientation (landscape).
  5. In the reference list, numbers are doubled in the document. Please revise.
  6. Some citations do not include DOI information (For example ref. 48, 63, 64,70). Please revise and unify.

Author Response

Journal: Nanomaterials

Manuscript ID: nanomaterials-1389863
Title: Nanotechnology innovations to enhance the therapeutic efficacy of quercetin

Reviewers' comments:

Referee 1:

The review entitled “ Nanotechnology innovations to enhance the therapeutic efficacy of quercetin” from Rúben G. R. Pinheiro et al. is a well-structured review focused in the nanocarriers which have been recently developed in order to improve quercetin therapeutic efficacy, increasing their solubility as well as their tissue-specific delivery.

In my opinion, the review would gain in interest if the authors could provide/discuss/correct the following aspects:

We acknowledge and appreciate the reviewer’s comments and we thank the reviewer for their opinion about our manuscript. We agree with the reviewer’s suggestions and the manuscript was carefully revised accordingly. The modifications proposed were highlighted in the manuscript. We believe that the observations added to the text are now in agreement with the considerations made. A more detailed explanation of the alterations made is subsequently described.

  1. In the introduction, authors should contextualize the review in the field of nanotechnology, citing other previous similar reviews (For example: DOI: 10.1039/c5ra18896b or DOI:10.1039/D1FO00851J) and including a brief sentence justifying the necessity of the present revision.

Thank you for you comment. In fact your suggestion is extremely relevant. In that sense we reformulated the manuscript, toutching in the introduction the aspects that you mentioned in your comment, as follows:

“1.1 Quercetin as a promising compound

Quercetin (3,3´,4´,5,7-pentahydroxyflavone) is a flavonol isolated and identified for the first time by Szent-Gyorgyi in 1936 (Figure 1) [1]. In the past, and similarly to what happened with other flavonoids, quercetin was classified as a vitamin showing to be important in maintaining capillary wall integrity and capillary resistance, antihypertensive and antiarrhythmic activity. In subsequent studies, it was proven to have anti-inflammatory, and antiallergic properties, hypocholesterolemic activity, platelet and mast cell stabilization, antihepatotoxic activity and antitumor action [2,3]. Lastly, more in depth analysis demonstrated that the beneficial effects of quercetin were related with its antioxidant (free-radical scavenging) and an-ti-inflammatory activity [4]. Additionally, some reports also attest its neuroprotective properties. Thus, quercetin emerged as a promising candidate for the treatment of many diseases such as cancer and neurodegenerative diseases. Nevertheless, in order to explore the potentialities of this flavanol, it is necessary to increase its bioavailability that is very reduced due to its hydrophobicity.

Nanotechnology is an appealing option to overcome this limitation since it can increase quercetin bioavailability at the desired site of action and, consequently, improve the therapeutic effectiveness. Hence, various kinds of nanoparticles, namely, liposomes, lipid nanoparticles, polymeric nanoparticles, cyclodextrins, silica nanoparticles, magnetic nanoparticles and gold nanoparticles, have been used to tackle this issue. Despite some authors have covered specific aspects of quercetin delivery such as a particular class of nanosystems (polymeric vehicles) or medical applications (tumor therapy) [5,6],, a comprehensive review summarizing all the data available regarding nanovehicles used for quercetin delivery is still needed. Intending to fill this gap, we aim to summarize the current literature regarding quercetin (e.g., chemical structure, natural sources, biosynthesis, pharmacokinetics and health benefits) and compile an extensive analysis of how nanotechnology have been employed to maximize the therapeutic potential of this flavanol in its wide range of medical applications.”

  1. The section “4. Nanoparticles for quercetin delivery” must include a sub-section about the “Biopolymeric nanoparticles”, as good candidates for drug delivery in medicinal applications because they are constituted by highly versatile, non-toxic, biocompatible and biodegradable molecules, with an increase number of examples and outstanding applications. There are many examples in literature describing nanoformulations of quercetin with silk fibroin (10.1016/j.ijpharm.2016.12.046; 10.1016/j.ijpharm.2021.120935), zein (doi: 10.3389/fchem.2020.00662), or chitosan (3390/polym10030235) among others.

We thank the reviewer for their suggestion which greatly improved this work. A new section was added devoted to the Biopolymeric Nanoparticles and Table 2 was also updated to include some examples using biopolymers, as follows:

“4.3.1 Biopolymeric nanoparticles

Interestingly, the sub-class of biopolymers are particularly attractive since they are non-toxic, biodegradable and biocompatible. This feature is majorly attributed to its monomer composition consisting essentially in nucleic acids, amino acids and saccharides from sugars. Indeed, there are reports in the literature describing biopolymeric nanoparticles to deliver quercetin. In one of these studies, Wang et al. developed zein and dextran sulfate sodium binary complex to deliver quercetin. This nanovehicle exhibited almost no-toxicity as well as a sustained released making it a promising candidate for future applications [125]. In another study Lozano-Pérez et al. synthesized silk fibroin nanoparticles loaded with quercetin and observed a controlled released under the conditions of pH characteristic of intestinal fluid which makes them suitable for gastrointestinal delivery [105]. In a follow up study, quercetin-loaded silk fibroin nanoparticles administered in a model mouse of colitis improved the pathological inflammation and preserved the normal crypt architecture, confirming the previously reported capacity for gastrointestinal delivery [106]. Another interesting biopolymer is chitosan, a cationic polysaccharide well known by its biological properties making it a good candidate when developing drug delivery systems [126]. Pedro et al. developed pH-responsive amphiphilic chitosan nanoparticles to encapsulate quercetin for a sustained release in cancer therapy. The results indicated a good hemocompatibility of the nanoparticles and suggested they could inhibit cancer cell growth in vitro. Therefore, these chitosan nanoparticles may help in reducing the side effects upon systematic administration of quercetin as anti-cancer therapy [107].”

Table 2. Properties of different quercetin-loaded carriers.

NPs Type

Composition

Morphology

Particle size (nm)

Zeta potential (mV)

Encapsulation efficieny (EE) (%)

Administration route

In vitro/in vivo results

Refs

Biopolymeric nanoparticles

Zein-Dextran sulfate sodium; Skin Fibroin; Chitosan

spherical

140 to 300

-16 to -39;
+14 to +30

20 to 78

oral, intravenous

- promising favorable and sustained delivery of quercetin (in vitro);

- controlled release in the pH of intestinal fluid; suitable for gastrointestinal delivery (in vitro);

- improving pathological inflammation and preserving normal crypt architecture in model mouse of colitis (in vivo);

- pH-responsive sustained release for cancer therapy; reducing side effects upon systematic administration (in vitro).

[104-107]

  1. The review presents a lack of schemes or figures presenting the wide variety of nanocarriers available for Quercetin transport and it would help to understand the different mechanisms of transport. Indeed, the figure 2 presents a lower resolution than figure 1. Please, check it.

Thank you for your comment. We have addressed and the quality of the figure 2 was improved.

  1. Please, edit the whole document according the journal requirements. The manuscript contain paragraphs with different text adjustment (for example in page 6) and check the page orientation (landscape).

Thank you for your comment. We have addressed this formating issue and all the manuscript was revised.

  1. In the reference list, numbers are doubled in the document. Please revise.

Thank you for your comment. We have addressed this formating issue and have deleted all doubled numbers in the reference list.

  1. Some citations do not include DOI information (For example ref. 48, 63, 64,70). Please revise and unify.

Thank you for your comment. We have introduced the DOI information in all available references lacking that information. We wanted to mention that in some cases we were not able to find a DOI, therefore these ones were left empty.

Reviewer 2 Report

The manuscript titled "Nanotechnology innovations to enhance the therapeutic efficacy of quercetin" provides a review of the structure/biosynthesis, sources, health benefits/toxicity, and delivery technology currently under study for quercetin.  The manuscript is well organized and contains sufficient information for anyone looking for updated information about current quercetin delivery nanotechnology.  I find the article to be suitable for acceptance with minor edits and do not have any specific suggestions or changes to request.

Author Response

The manuscript titled "Nanotechnology innovations to enhance the therapeutic efficacy of quercetin" provides a review of the structure/biosynthesis, sources, health benefits/toxicity, and delivery technology currently under study for quercetin.  The manuscript is well organized and contains sufficient information for anyone looking for updated information about current quercetin delivery nanotechnology.  I find the article to be suitable for acceptance with minor edits and do not have any specific suggestions or changes to request.

We acknowledge and appreciate the reviewer’s comments and we thank the reviewer for the positive opinion about our manuscript.

Reviewer 3 Report

In this well written review Pinheiro et al. provide a comprehensive and complete picture of the current field of nanoparticle-based approaches to enhance the therapeutic efficacy of quercetin.  Maybe an additional paragraph on the limitations of those nanotechnology-based strategies would be welcome in order to support further progress.

Overall, the review is well organized and clearly written. The tables support the text properly, but I have noticed some points which should be improved.

In the Table 2, the section of <in vitro/in vivo results> includes both results of in vitro and in vivo. The authors tried to express which results are from in vitro or in vivo. However, some is confusing. Please express them more clearly.

In the Table 2, what does brain mean as an administration route of lipid nanoparticles?

In the Table 2, what does EE (%) mean?

In the Table 3, the contents of intervention are confusing. What is difference between quercetin and drug: quercetin? If possible, it would be helpful to express of administration route and formulation of quercetin.

In the Table 3, many clinical trials show <There is no results> or <not indicated> even though they are completed. I recommend to discuss why they have no results in the text.

I found a spelling mistake (HOMO).

Author Response

In this well written review Pinheiro et al. provide a comprehensive and complete picture of the current field of nanoparticle-based approaches to enhance the therapeutic efficacy of quercetin.  Maybe an additional paragraph on the limitations of those nanotechnology-based strategies would be welcome in order to support further progress.

We acknowledge the referee comment and we completely agree that this discussion can greatly improve the manuscript. Following the suggestion of the reviewer, the Conclusion Section was carefully revised in order to accomplish the limitations of the current review, as follows:

“Thinking on the beneficial effects of quercetin, a wide range of strategies that use nanoparticles have been developed to deliver quercetin in a specific and controlled way. The revised nanosystems described in the literature revealed promising approaches with a great capacity for quercetin encapsulation and controlled release. Moreover, several strategies of nanoparticle functionalization have allowed the tissue-specific delivery of quercetin to tumor microenvironments as well as to increase its blood brain barrier permeation or the penetration within skin layers. However, there is a lack of clinical trials employing nanodelivery systems for quercetin administration, considering all the advantages mentioned in this review. Therefore, we identify here a potential area to be explored in future reviews, trying to understand why such promising nanosystems are not yet being widely used in clinical practice and how we can enhance their future use by taking advantage of the therapeutic properties of such a promising compound – quercetin. This strategy has great potential to be explored in a short period of time. Hence, the beneficial effects of quercetin may be further enhanced using nanotechnology, thereby helping to improve the current application of this compound with so great therapeutic potential.

Overall, the review is well organized and clearly written. The tables support the text properly, but I have noticed some points which should be improved.

In the Table 2, the section of <in vitro/in vivo results> includes both results of in vitro and in vivo. The authors tried to express which results are from in vitro or in vivo. However, some is confusing. Please express them more clearly.

Thank you for the comment. According to the reviewer suggestion, we have indicated which results were obtained from in vitro or in vivo experiments. This greatly helps to clarify the readers.

In the Table 2, what does brain mean as an administration route of lipid nanoparticles?

Thank you for your comment. It was a mistake we already corrected the Table 2.

In the Table 2, what does EE (%) mean?

Thank you for your comment. EE stands for encapsulation efficiency. We already added this designation to Table 2.

In the Table 3, the contents of intervention are confusing. What is difference between quercetin and drug: quercetin? If possible, it would be helpful to express of administration route and formulation of quercetin.

We understand the reviewer’s point. However we decided to use the nomenclature that the authors provided on the http://clinicaltrials.gov website in order to be consistent with the source information and to make easier for someone interested to search for the respective clinical trial. Please be advised that we are not always able to find the complete information regarding each clinical trial that has been developed, as is the case of the administration route or the formulation of quercetin. So, we decided not to include this information only for some cases, in order to have consistency in the information provided for all clinical trials.

In the Table 3, many clinical trials show <There is no results> or <not indicated> even though they are completed. I recommend to discuss why they have no results in the text.

Thank you for your comment. We already included in the text a speculative hypothesis for the lack of results, as follows: “Although many studies have been completed, only a few have published the results, perhaps due to inconclusive results”.

I found a spelling mistake (HOMO).

Thank you for your comment. We already corrected this typo as follows: “HOMO (highest occupied molecular orbital)”.
